# Get Rid of Marine Pollution: Bioremediation an Innovative, Attractive, and Successful Cleaning Strategy

Valbona Aliko [1], Cristiana Roberta Multisanti [2], Blerta Turani [3] and Caterina Faggio [2,*]

1   Department of Biology, Faculty of Natural Science, University of Tirana, Boulevard "Zogu I", 25/1,
    1001 Tirana, Albania
2   Department of Chemical, Biological, Pharmaceutical, and Environmental Sciences, University of Messina,
    Viale Ferdinando Stagno d'Alcontres, 31, 98166 Messina, Italy
3   Department of Food, Technology, High Professional College, "Qirjazi" University, 1029 Tirana, Albania
*   Correspondence: cfaggio@unime.it

**Abstract:** Aquatic environmental pollution is a rather worrying and increasingly topical problem that requires the development and promotion of innovative and ecofriendly technologies. Pollutants in water include many common substances that can reach aquatic ecosystems through several pathways including wastewater, the atmosphere, ship discharges, and many other sources. Most of these toxic compounds are internalized by aquatic organisms, leading to bioaccumulation in tissues and reaching any level of the food chain through the biomagnification process. These mechanisms can develop into adverse effects on the physiology of organisms and biochemical processes of natural ecosystems, thus affecting animals, environments, and indirectly, human health. Innovative technologies to tackle marine pollution include bioremediation: a suitable, biological, and ecological approach that enhances the ability of micro-organisms to transform waste and toxic substances into forms that can be used by other organisms. In this context, micro-organisms appear to be essential for the detoxification of aquatic systems due to their metabolic activity. This review provides a careful analysis of the characteristics of the main pollutants that affect aquatic ecosystems, with a focus on their effects on organisms and environments. It also offers clear guidance on innovative biological strategies that can be employed to prevent, limit, and remediate anthropogenic influences on aquatic environments.

**Keywords:** anthropic disturbances; marine biology; innovative tools; contaminants; aquatic pollution

## 1. Introduction

The expression "marine pollution" refers to an environment in which the occurrence of several micropollutants, such as chemical and/or biological substances and wastes, carried by various sources, but mainly terrestrial, may lead to interferences at different biological levels.

Nowadays, our seawater has become a major recipient of chemicals from all around the world. Indeed, remote areas of the earth are now besieged by the presence of toxic substances, and the atmosphere, adverse weather conditions, and sewage treatment plants which are not always efficient play a key role in the dispersion and deposition of many compounds over long distances from their source. Currently, not all these pollutants have been well-studied, but some have been banned or legislated against based on their toxic effect on the environment. Emerging pollutants include a wide range of different compounds: personal care products, pharmaceuticals, new pesticides, industrial compounds, debris, micro/nanoplastics, and even toxic biomatter [1–4]. Some of these compounds act by interfering with the endocrine system, mimicking or hindering normal hormonal action in organisms, including humans, and impairing numerous functions, such as growth, metabolism, and reproduction [5,6].

Three different sources of pollution have been identified:

1. From Land.

Human activity on land is responsible for 80% of marine pollution [7]. Indeed, the major input of toxic substances into marine water comes from pipelines that discharge pollutants of all kinds. However, river transport must also be considered because it allows the transport of toxic substances from the entire catchment area to the sea.

2. From Air.

The atmosphere plays a crucial role as global toxic inputs to the sea are due to atmospheric discharges.

3. From Water.

Oily discharges are released daily through ballast and bilge waters due to the illegal dumping of waste from ships. Unfortunately, it often happens that toxic substances are also accidentally lost from ships. In addition, pollutants are continuously released into the sea from other sources (dredging materials, sewage sludge, fly ash, and oil-based sludge).

Aquatic pollution and its incidence worldwide is increasing day by day, and this growing problem needs new control and monitoring strategies. A high percentage of species suffer from anthropogenic effects in aquatic systems. For example, many specimens of the common marine turtles *Caretta caretta* beached themselves on Sicilian beaches due to their negative health conditions as a result of anthropic activities [8], and *Bufo bufo* tadpoles exposed to environmentally realistic doses of fluoxetine and ibuprofen showed impairment in their development and fitness [9].

Various wastes can be mistaken as food by marine mammals, fish, and seabirds, with disastrous and sometimes fatal effects.

To assess aquatic pollution levels, a careful investigation based on the measurement of abiotic components is required. In addition, bioaccumulation in specific organisms, selected based on their characteristics, must also be considered [10]. Several organisms, vertebrates, and invertebrates are commonly used as bioindicators because of their ability to accumulate high levels of substances in their tissues [11,12]. In a polluted environment, organisms can be continuously exposed to low yet continuous levels of environmental pollutants despite dilution by water masses. It has been widely established that, even at low concentrations, these exposures negatively interfere with the entire ecosystem [13].

## 2. Toxic Substances and Their Effects

### 2.1. Heavy Metals

A high percentage of pollutants, such as heavy metals, is continually released into the marine environment through human activities, which represents a very critical issue [14–17]. Anthropogenic sources of metals include urban runoff, sewage, traffic emissions, coal and oil combustion, industrial production, mining, and the smelting of ores [18]. Marine organisms ingest heavy metal ions from their diet, and exposure to metals at above-threshold concentrations is extremely toxic [19]. Several xenobiotics, including heavy metals [16,20–25], can have long-term effects, which are not immediately visible, that involve alterations in molecular and cellular responses, and these appear to have a major impact on ecosystems.

Currently, pollution caused by heavy metals as a result of man-made activities is becoming a real concern for aquatic animals. The elimination of heavy metals at the tissue level is not constant. Indeed, several factors come into play, such as the time of exposure, temperature, interacting agents, the metabolic activity of the animal, and the chemical composition of the metal [26]. The lack of detoxification and the accumulation of heavy metals in fish can lead to physiological and pathological alterations in tissues and organs [27]. Oxidative stress is one of the most investigated effects as exposure to heavy metals causes the production of reactive oxygen species (ROS). Recent data showed that the exposure of Mozambique tilapia (*Oreochromis mossambicus*) to selenium resulted in the alteration in natural antioxidant enzyme activity with an increase in superoxide dismutase (SOD), glutathione peroxidase (GPx), glutathione reductase (GSH), metallothioneins (MT), and catalase inhibition (CAT) in the gills and liver [28], thus indicating high levels of stress in the

fish. In invertebrate organisms, these substances show effects on the central nervous system and physiology as exposure to pollution-induced stress affects their homeostasis [29].

### 2.2. Microplastics and Nanoplastics

Micro/nanoplastic particles have primary and secondary sources. Primary microplastics come from industrial and domestic sources including personal hygiene products, such as shampoos, shower gels, and toothpaste as well as laundry fibers. Secondary microplastics result from the breakdown of macroscopic oceanic plastic debris [3,30]. Secondary microplastics include waste, which seems to be the main source of secondary microplastics, fibers, and plastic material from organic matter [31].

Long-term exposure to agents such as physical abrasion and ultraviolet light leads to the fragmentation of plastic objects into smaller units [32], namely, micro/nanoplastic particles, for which degradation is not easy. Based on their dimensions, plastic particles can be classified, as shown in Table 1.

**Table 1.** Classification of plastic debris based on their size according to GESAMP [33].

| Definition | Size–Range | Description |
|---|---|---|
| Nanoplastics | >100 nm | The smallest plastic particles which can only be observed under scanning electron microscopy (SEM) and transmission electron microscopy (TEM). |
| Microplastics | 100 nm–1 mm | Small plastic particles which can be visualized under light microscopy. |
| Mesoplastics | 1 mm–2.5 cm | This term was introduced to distinguish plastic particles that can be observed by the naked eye, unlike the previous two categories. |

Among the different types of microplastic particles, microfibers consisting of nylon, polyethylene terephthalate, and polypropylene are continuously released into aquatic systems mainly from the textile industry and from synthetic clothes washed in washing machines. Microfibers can negatively interfere on several levels. The gradual increase in these types of pollutants will result in a covering of the ocean surface with a layer of microfibers that will decrease the level of oxygen in the water. The microfiber particles ingested by different marine organisms lead to numerous types of damage, including reduced feeding capacity, reproductive abnormalities, liver toxicity, and decreased reproductive potential [34,35]. Fiber particles are often associated with several chemical constituents that are potentially toxic to aquatic systems as well as to humans such as phthalates, bisphenol A (BPA), formaldehyde, and Teflon [36].

Recent studies have also reported that microfibers have been found in different species that are commonly consumed by humans: the Mediterranean green crab (*Carcinus aestuarii*), the lady crab (*Callinectes sapidus*), the blue mussel (*Mytilus edulis*), and the Mediterranean mussel (*M. galloprovincialis*) [36]. As reported later in this paper, these species are commonly used as bioindicators to test water quality. Specimens of *Holothuria tubulosa*, often used as bioindicators, can also be affected by microplastics. In particular, it was observed that microplastic polymers can alter the expression and activity of enzymes involved in oxidative stress [37].

Even though there is evidence that micro/nanoplastics negatively affect the development and fitness of nontarget organisms, it is important to understand that the impact of micro/nanoplastics needs further in-depth research [38].

### 2.3. Pesticides

Pesticides are particularly used in agriculture and may also interfere biologically and ecologically with aquatic organisms. These compounds can induce changes in behaviour and impair chances of survival [39,40].

Herbicides, a subcategory of pesticides, are commonly used to control algal growth. Uncontrolled algae growth can impede water flow in summer when sudden heavy rain causes flooding. These substances aim to reduce macrophytes, but they also affect nontarget organisms that are subjected to the loss of their habitat and food [41]. The presence of these excess nutrients can generate massive algal blooms that deprive water of the oxygen necessary for marine life [42,43].

Insecticides are also potentially harmful to marine ecosystems because they are easily transported through the atmosphere, sewage treatment plants, sewers that may overflow during rainy seasons, and accidental leaks of insecticides on farms or near areas where they are widely used. In these cases, the release of insecticides into irrigation ponds unequipped with appropriate safety devices results in their direct release. Nevertheless, the major source is surface water runoff [44]. These compounds are particularly used in agriculture and are potentially toxic to aquatic organisms as well as to human health. According to Stara et al. [45] and Barathinivas et al. [46], they can compromise physiological processes such as cell viability and hemocyte biochemical parameters in nontarget organisms, leading to oxidative stress.

### 2.4. Common Compounds inside Personal Care Products

Other substances that can potentially have a toxic effect on the environment are components of Personal Care Products, which are now used daily [3,47]. These products contain antibacterial and antifungal chemicals, and by entering water bodies, they can accumulate worldwide [48]. Triclosan, an antimicrobial compound found in soaps, deodorants, and gel showers, is one of the most common organic micropollutants in aquatic systems worldwide. It can be found in wastewater treatment plant effluents, surface water, and sediments due to its abundant use and incomplete removal by wastewater treatment plants [49], with a mean concentration between 0.0075 μg/L and 9.65 μg/L [50]. According to Dar et al. [51], triclosan is responsible for the paralysis of many fish larvae, such as *Cyprinus carpio*, *Ctenopharyngodon idella*, *Labeo rohita*, and *Cirrhinus mrigala*, probably as a result of biochemical and transcriptomic alterations leading to oxidative stress, abnormalities in the normal function of the kidney and digestive system, and the impairment of normal metabolic processes [52].

The evidence shows that UV filters, which are components of sunscreens, are among the compounds mainly contained in aquatic systems to date. These compounds can enter aquatic environments through recreational human activities (such as swimming). Due to their stability and lipophobicity, UV filters have been shown to be particularly persistent in aquatic ecosystems and toxic to nontarget organisms. Their toxicity tends to bioaccumulate in fish tissue and acts as an endocrine disruptor. Toxic compounds that accumulate in the tissues of aquatic organisms can interfere with the endocrine system and are, therefore, defined as persistent, lipophilic, and bioaccumulative. Estrogens and other endocrine disruptors may also be present in water bodies because wastewater treatment plants are unable to completely remove these hormones. They are called "endocrine disruptors" because these substances affect the synthesis, secretion, transport, binding, action, and elimination of natural hormones that play a key role in maintaining homeostasis, reproduction, development, and behaviour [53]. Humans can also come into contact with endocrine disruptors. This happens if untreated groundwater is used as drinking water, and even bottled water can contain them due to the plasticizers and detergents employed during their fabrication.

Critical toxic effects were observed from the exposure of zebrafish embryos to an environmentally relevant concentration of a specific UV filter, BP-3 (10 μg/L). BP-3 appears to be the most toxic of the UV filters. Indeed, it causes a stressful condition in zebrafish,

resulting in increased spontaneous movements and decreased axis growth in embryos, increased hyperactivity, decreased shoaling behavior, and a decrease in cellular proliferation in the larvae [54].

The evidence on the toxicological and apoptosis-inducer potential of environmental realistic doses of fluoxetine (FLX) and ibuprofen (IBU) on tadpoles of the common bullfrog, *Bufo bufo*, have also been reported [9]. Both drugs displayed a negative effect on tadpole development as well as erythrocyte morphology and behaviour, compromising the overall fitness of the species and their population dynamics.

### 2.5. The Combined Action of Pollutants

An aspect of pollution that has not yet been fully investigated is how the combined action of different toxicants can affect organisms and ecosystems. For example, problems caused by the simultaneous action of organochlorine pesticides and polychlorinated biphenyls (PCBs), a group of toxic substances classified by their chemical and biological stability, have not been studied a lot in freshwater ecosystems. It was observed that in crucian carp (*Carassius carassius*), the synergistic action of the two toxicants leads to kidney alterations [17].

The condition of hypoxia due to anthropogenic pollution also leads to serious problems in freshwater, estuarine, and marine systems worldwide. The combined action of hypoxia, PCBs, and pesticides leads to significant increases in blood glucose (GLU), cortisol levels, hematocrit (PCV), and hemoglobin (Hb) in the organism, which indicates stress conditions in fish [33].

### 2.6. Contamination Due to Biological Pollutants

Biological contaminants include several micro-organisms that, if ingested, can interfere with the organism of other life forms. The most common mode of transmission of these organisms is the fecal–oral pathway.

Major biological contaminants include:

1. Pathogenic bacteria;
2. Coliforms;
3. Fecal streptococci;
4. Clostridium perfringens;
5. Viruses;
6. Protozoa;
7. Helminths.

All these pollutants are easily transported by water.

Algal blooms [55] and the growth of sewer fungi can also be considered forms of aquatic pollution. Blue-green algae are among the most obvious examples of this type of pollution. They have the ability to produce novel toxins for terrestrial organisms that also impart unpleasant tastes and smells to the water.

Noise pollution must also be considered. Loud or persistent sounds can interfere with the migration, communication, hunting, and reproduction patterns of many aquatic organisms, especially in the case of mammals such as cetaceans and dolphins. The main sources of these disturbances are ships, sonars, oil platforms, and even natural sources such as earthquakes [56,57].

The decline in the abundance of large marine fauna is the direct responsibility of the impact of humans that has intensified since the 20th century. Physiological and pathological responses to stressors play a key role in enabling animals to cope with environmental perturbations but are poorly characterized in marine mammals. Indeed, the anthropogenic stressors to which marine mammals are subjected include exposure to pathogens, pollution, and noise.

Each of these pollutants has chemical characteristics that underlie the effects they have on organisms and ecosystems. Therefore, increasingly specific investigations are essential to fully understand toxic molecule–organism interactions. The results obtained

to date provide a good overview of the toxic effects of the main compounds present in aquatic systems, providing the opportunity to predict and implement the most effective and up-to-date prevention and degradation methods.

## 3. Bioaccumulation, Bioconcentration, and Biomagnification

In aquatic systems, the two relevant processes are "bioaccumulation" and "biomagnification", which, although different, often occur in parallel.

The term bioaccumulation refers to the ingestion through water and food and the subsequent accumulation of toxic substances in the tissues of individual organisms, while biomagnification allows the transfer of toxins and their amplification of concentration from one trophic level to the next. During bioaccumulation, some of these compounds can be degraded by normal digestive processes into nonharmful substances, while others that cannot be broken down remain within the exposed organism. In this context, investigations into the capability of the bioaccumulation of heavy metals are essential. Heavy metals, such as Cu and Zn, are the essential cofactors of biochemical enzymatic reactions for the organism, but excessive amounts lead to life-threatening damage. For this reason, it is important to assess their concentration in surface waters and their bioaccumulation capacity using suitable bioindicators, such as *M. galloprovincialis* [58].

When a specimen is preyed upon by another belonging to the next trophic level, the accumulated substances will be transferred up the food chain and retained in increasing concentrations through the process of biomagnification.

Bioconcentration is an active process which is defined as the accumulation of a chemical in an organism when the source is only water and is the process through which the concentration of the substance is internalized in an aquatic organism and overcomes the concentration found in water [59,60]. To measure and assess bioconcentration, a mathematics model can be employed: the bioconcentration factors (BCF) represent the concentration quotient of a pollutant in an organism related to its surrounding environment [61].

*Bioindicators*

The analysis of the growth of marine organisms is a relevant and sensitive index for assessing environmental stress. This is one of the endpoints to evaluate a suitable bioindicator.

For example, marine invertebrates are routinely used in the laboratory because of their ability to survive in polluted environments despite accumulating high levels of heavy metals. Metals penetrate cells in their tissues using common transport mechanisms and irreversibly accumulate, interacting with cellular components and molecular targets. Importantly, invertebrates occupy a key position in pelagic and benthic food chains as intermediate consumers. These include bivalves, a species well known in ecotoxicology for assessing the ecohealth of aquatic environments [12,62].

A suitable bioindicator species, such as the Mediterranean mussel (*Mytilus galloprovincialis*), should be used to provide accurate and reliable measurements of environmental quality. *M. galloprovincialis* is a sessile organism that lives in close contact with hard substrates, where its exposure and relative inability to move allow it to record surrounding changes. For all these reasons, it has been widely used to monitor the effects of pollutants in different ways, i.e., chemical analysis and biological responses.

In numerous studies, the same approach has been tested on the North Atlantic portunid crab (*Carcinus maenas*) and the Mediterranean green crab (*Carcinus aestuarii*) [63]. These species are sensitive to many pollutants and are, therefore, a reliable model for routine testing in ecotoxicological research and water quality assessment [64].

Fish are now widely used as bioindicators in water quality monitoring due to their ability to respond to changes in aquatic environments with great sensitivity [17,65,66]. There are numerous endogenous and exogenous factors involved in the modulation of hematological parameters in fish. The alteration of these factors induces stress and physiological abnormalities [67].

Fish can be directly and indirectly influenced by contaminants. Direct effects impact the lower levels of biological organization, while indirect effects, on the other hand, involve the food chain and affect the behaviour of organisms [68].

Due to their characteristics, invertebrate organisms are probably the most suitable for environmental biomonitoring studies. In particular, bivalve mollusks represent an excellent model organism for biomonitoring the surrounding environment due to their feeding mode (filter-feeding) and their constant occurrence in time and space.

## 4. Bioremediation of Marine Ecosystems

Over the last few decades, the increasing anthropogenic impact on marine ecosystems has become an issue of concern worldwide. The increasing pressure on natural resources due to population growth and pollution also harms water and land resources. Polluted coastal and marine environments are the result of the continuous neglect and negligence of human activities on both marine and terrestrial natural resources.

In this regard, several effective bioremediation strategies have been developed. However, bioremediation techniques have to be compatible with the major natural biogeochemical cycles and recycling pathways of terrestrial and marine ecosystems in order for them to constitute ecofriendly approaches for the remediation of environments [69].

Bioremediation is an emerging technique in environmental biotechnology that employs the specific metabolic activities of bacteria, fungi, yeast, microalgae, and microbial mats to purify ecosystems. The activity of these micro-organisms is crucial as they are able to mineralize or transform organic contaminants into less harmful substances.

This is an economical and above all, nondestructive treatment, as shown in Figure 1.

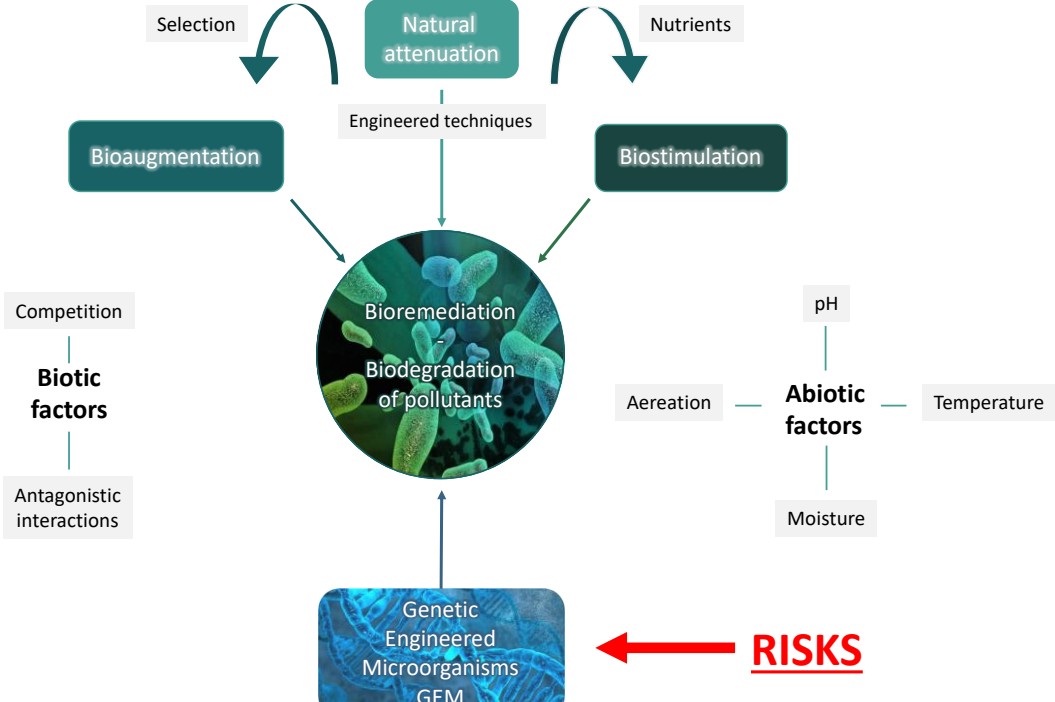

**Figure 1.** Bioremediation of pollutants using biodegradation carried out by micro-organisms includes natural attenuation. This can be further enhanced using engineering techniques through the addition of micro-organisms selected according to two processes called "bioaugmentation" or "biostimulation" in which nutrients are added. Genetic engineering through GEM can also be used to enhance the action of micro-organisms. Unfortunately, there are a lot of risks associated with the efficiency of this process.

Bioremediation follows two main strategies: biostimulation and bioaugmentation. The former involves the stimulation of indigenous microbial populations, while the latter involves the introduction of viable microbial populations. Either route can be chosen based on an analysis of the abiotic and biotic factors that influence the biodegradation process. The growth of bioremediation has led to the production of useful spill cleanup products, such as fertilizers with biostimulating nutrients, micro-organism-based bioproducts, and chemicals to stimulate the growth of microbial populations [69].

Marine environments differ based on numerous parameters that affect them: temperature, pH, salinity, currents, precipitation, and winds. Associated with these chemical differences, there is a high diversity of specific micro-organisms, which is important due to the roles they play. They quickly respond with high sensitivity to surrounding environmental changes, and this property makes them ideal subjects for bioremediation processes. Substances that can be removed by micro-organisms isolated from marine environments include heavy metals, hydrocarbons, and many other recalcitrant compounds.

Nowadays, bioremediation is a process of increasing interest because of its efficiency and reliability but, although the rapid adaptive capacity of micro-organisms to environmental changes is confirmed, little is known about their resistance to harmful environments.

A good example of bioremediation is the natural degradation of oil. Often oil is removed over time by natural processes, but in the case of larger losses, human intervention is required with the knowledge of specific processes whereby the residual oil can be broken down using an artificial method called biostimulation [70]. This is discussed briefly below.

### 4.1. Biostimulation

Biostimulation is an innovative process which provides benefits through the addition of specific nutrients such as air, organic substrates, or other electron donors/acceptors. These nutrients speed up the bioremediation process and make it more efficient.

This is an excellent mechanism when the bacteria needed to degrade a specific type of waste is naturally found in the environment to be treated.

### 4.2. Bioaugmentation

Bioaugmentation comes into play when you want to control, predict, and plan for biodegradation. Specifically, it is the controlled addition of highly specialized microbial cultures to assist populations that already exist in nature. The cultures are achieved using a favorable growth environment for these specific bacteria in which they are able to work [71].

When performing bioaugmentation, it is important and necessary to monitor certain elements that, if left unchecked, can lead to a failed outcome [72]:

1. Check that the substrate concentration is sufficient to support the growth of the microbial population;
2. Be certain that the system does not have any components that may inhibit the process, such as temperature;
3. Competition with other micro-organisms causes growth inhibition;
4. Inoculated micro-organisms may degrade other organics rather than the target pollutant substrate;
5. Make sure the number of micro-organisms is sufficient for the process;
6. Use organisms specific to the type of substrate to be degraded.

If all the elements listed above are not a problem, then bioaugmentation is an excellent strategic opportunity.

## 5. Pathways for Bioremediation of Marine Ecosystems

### 5.1. Biodegradation of Petroleum Hydrocarbons

One of the most efficient mechanisms to remove oil-polluting hydrocarbons from the environment is their degradation using micro-organisms. The micro-organisms capable of this specific degradation are bacteria, fungi, and yeasts of different species such as *Arthrobacter, Burkholderia, Mycobacterium, Pseudomonas, Sphingomonas*, and *Rhodococcus* which, according to a study by Jones et al. [73], are responsible for the degradation of alkyl aromatics in marine sediments. According to the authors Jahangeer and Vikram Kumar [74], a single population cannot degrade complex mixtures of hydrocarbons in aquatic environments, but a mixed population rich in enzymatic properties certainly is capable of this.

Recently, microbial communities capable of cometabolizing toxic chemicals have been identified and used. These include rhizosphere bacteria [75] such as the *Bacillus* species for hydrocarbons [76] and the *Paraburkholderia* species for aromatic compounds [77].

For most organic pollutants, the aerobic one is the best condition for biodegradation to occur. The enzymes mainly involved in the intracellular attachment of inorganic pollutants and the activation and incorporation of $O_2$ are oxygenase and peroxidase, as shown in Figure 2.

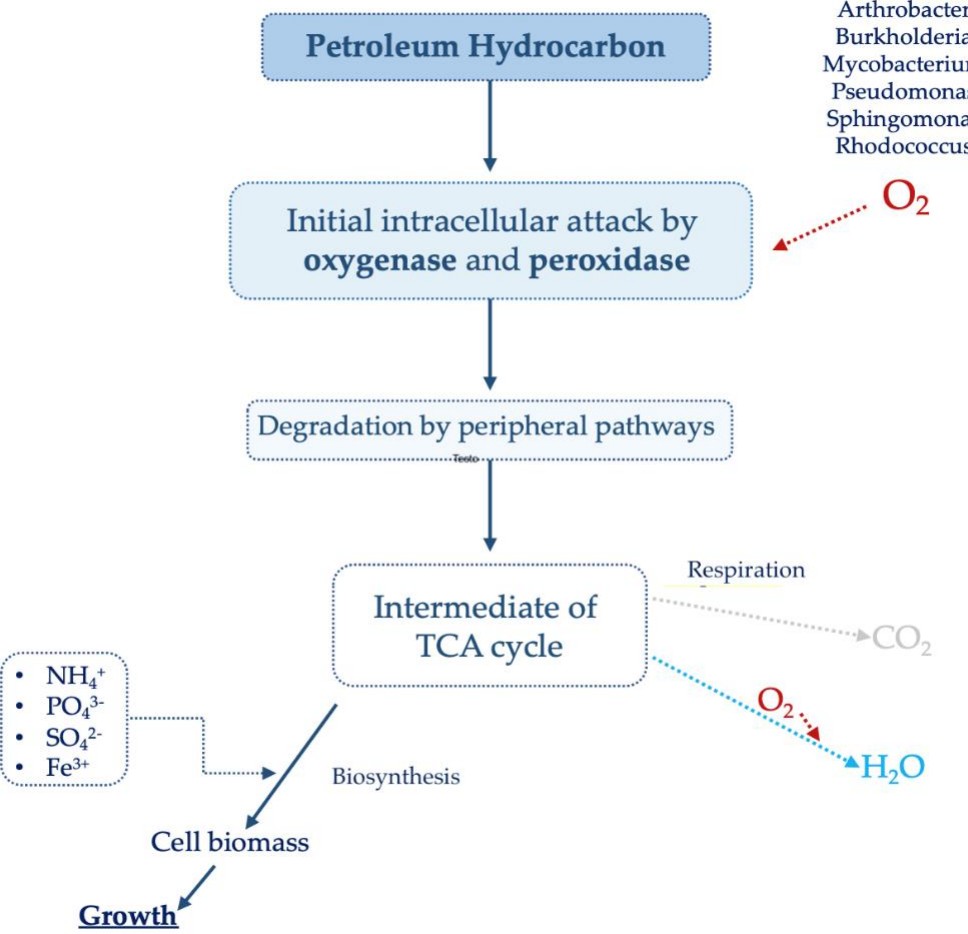

**Figure 2.** Schematic representation of the main steps of microbiological activity for petroleum hydrocarbon biodegradation.

Secondary degradation pathways allow the conversion of pollutants into metabolic intermediates, such as the tricarboxylic acid cycle. The biosynthesis of cell biomass takes place from the central precursor metabolites, such as acetyl-CoA, succinate, and pyruvate. In addition, the essential sugars required for biosynthetic pathways are synthesized through

gluconeogenesis. Cytochrome P450 alkane hydroxylases are a large family of ubiquitous hemethylated monooxygenases crucial for the microbial degradation of oil and other compounds. Indeed, some yeast species, such as *Candida maltosa*, *Candida tropicalis*, and *Candida apicola,* are able to utilize n-alkanes and other aliphatic hydrocarbons as a source of carbon, hence energy, due to their microsomal multiple forms of cytochrome P-450.

Industrial production, waste combustion, gasification, and incineration of plastics are the main contributors to polycyclic aromatic hydrocarbon (PAHs) pollution, especially in recent decades. PAHs are hydrophobic substances that can easily adsorb particulate matter and as a result, coastal and marine sediments become their sinks. Low molecular mass PAHs, such as naphthalene, phenanthrene, and anthracene, have been particularly targeted. Micro-organisms capable of degrading petroleum hydrocarbons are normally low in abundance in marine environments, but if present, they can stimulate their growth. To understand and evaluate strategies to bioremediate an area, it is necessary to know and identify the organisms that can carry out this process [78].

To make the bioremediation of PAHs more efficient, new bacterial species present in water such as Novosphingobium pentaromativorans US6-1, Cycloclasticus spirillensus, *Lutibacterium anuloederans*, *Neptunomonas naphthovorans*, and *Vibrio cyclotrophicus* have been used [79]. However, other species such as *Achromobacter denitrificans*, *Bacillus cereus*, *Corynebacterium renale*, *Cyclotrophicus* sp., *Moraxella* sp., *Mycobacterium* sp., *Burkholderia cepacia*, *Pseudomonas fluorescens*, *Pseudomonas paucimobilis*, *P. putida*, *Brevundimonas vesicular*, *Comamonas testosteroni*, *Rhodococcus* sp., *Streptomyces* sp., and *Vibrio* sp. were isolated and found to be involved in the degradation of naphthalene, one of the major PAHs, through its mineralization. *Pseudomonas* sp. and *Ochrobatrum* sp., two other species isolated from coastal sediments, are responsible for the degradation of fluorene and naphthalene.

To assess the metabolic potential of organisms capable of degrading PAHs, a strategy involving DNA–SIP labelling and metagenomic sequence analysis was deployed. Most of the reported sequences belong to *Betaproteobacteria*, specifically *Rhodocyclaceae* and *Burkholderiales* which are major PAH degraders. However, the problem remains that PAHs are more resistant to degradation, leading to a deleterious impact on marine and coastal sediment ecosystems [69].

### 5.2. Bioremediation of Heavy Metal Pollutants

Heavy metals have a deleterious impact on the diversity of micro-organisms in marine and coastal environments, which consequently leads to problems in ecosystem functioning as they cannot be easily removed. Recent studies have reported two sets of sequences associated with α-proteobacteria and actinobacteria, the micro-organisms found in metal-contaminated soils.

Marine bacteria such as *Enterobacter cloacae* are capable of chelating heavy metals through the secretion of exopolysaccharides. Other purple nonsulfur marine bacteria such as *Rhodobium marinum* and *Rhodobacter sphaeroides* have been studied and found to be capable of metabolizing heavy metals, such as Cu, Zn, Cd, and Pb, through bioadsorption. Current studies, therefore, aim to investigate metal–microbe interaction and its application for metal accumulation or detoxification [68].

### 5.3. Bioremediation of Marine Plastic Pollution

Several recent studies have shown that microbial degradation of plastic is an ecofriendly and efficient conversion. Several species of bacteria and fungi have been isolated that are capable of degrading different types of plastic polymers including *Penicillium* sp., *R. arrizus*, *R. delemar*, *Achromobac-ter* sp., *Candida cylindracea*, *Penicillium* sp., *Aspergillus* sp., *R. arrizus*, *Clostridium* sp., *Roseateles depolymerans*, *Amycolatopsis* sp., *Candida cylindracea*, *Pseudomonas* sp., *Chromo-bacterium viscous*, *R. arrhizus*, *R. delemar*, *Curvularia senegalensis*, *Comamonas aci-dovorans*, *Acinetobacter* sp., *R. ruber*, *R. eutropha*, *R. rubrum*, *Pseudomonas*, *Alcaligenes*, and *Thermobifida fusca*.

Their common property is the ability to break down complex polymers into smaller ones by employing depolymerase enzymes before they interact with the cell. Two main groups of enzymes have been highlighted: intracellular and extracellular depolymerase. The extracellular enzymes help to break down the plastic polymer into shorter, water-soluble chains, while these chains enter the microbial cell and are metabolized by the intracellular enzymes. Furthermore, different groups of depolymerases have been isolated from a large number of microbial species (both bacteria and fungi) that are capable of degrading different types of plastics.

In general, the biodegradation process involves four steps:

1. Biodeterioration;
2. Biofragmentation;
3. Assimilation;
4. Mineralization.

The "Biodeterioration" process involves the formation of a biofilm around the plastic polymer, indicating the beginning of the degradation process. This appears to be the most important stage because it allows micro-organisms to access the polymers for their hydrolytic activities. The second phase, "biofragmentation", involves the secretion of the first (extracellular) enzymes by the micro-organisms [80] that allows the degradation of the plastic polymer, preparing it for ingestion. The "assimilation" phase involves the assembly of the oligomer/dimer/monomer on the surface of the micro-organisms and uptake via two possible pathways: simple or facilitated diffusion. Finally, "mineralization" entails the production of secondary metabolites, such as $CO_2$, $H_2O$, and $CH_4$ [81].

*5.4. Bioremediation of Pesticides*

Pesticides can be degraded by micro-organisms using an active process that requires outside energy or a passive process that involves physicochemical interaction with the structures on the cell wall.

Several processes are involved in the removal of pesticides by microalgae: bioadsorption, bioaccumulation, and biodegradation [82].

Bioadsorption is a passive process [83] that comprises different mechanisms such as electrostatic interaction, surface complexation, ion exchange, absorption, and precipitation [84–86]. According to Hussein et al. [87], 87–96% of some pesticides in water can be removed via bioadsorption because their cell wall is composed of carbohydrates, a fibril matrix, intercellular spaces, and sulfated polysaccharides that facilitate the contaminant adsorption from water.

Exposure to organic contaminants stimulates the production of reactive oxygen species [88] and consequently, the expression of inducible genes in the microalgal cells to produce antioxidant enzymes that activate the detoxification protection mechanism of microalgae.

Finally, biodegradation is a necessary process that allows microalgae to degrade organic matter into small molecules. These will become nutrient sources for their growth [89], involving the main enzymes such as esterase, transferase, and cytochrome P450, and in addition, hydrolase, phosphatase, phosphodiesterase, oxygenase, and oxidoreductases [90].

*5.5. Genetic Manipulation in Marine Bacteria to Enhance Bioremediation Efficiency*

Anthropogenic pollution has impaired bioremediation by bacteria through the introduction of substances to which they have not previously been exposed. However, after initial exposure, bacteria have the ability to modify their metabolism to withstand induced stress and survive. Through genetic manipulation, it is possible to improve the bioremediation potential of bacteria and/or their metabolic activity. Thanks to these techniques, it is possible to introduce new genes and new plasmids into the bacterial genome, modify metabolic pathways such as transport and chemotaxis, and above all, enable adaptation to new environmental conditions [91]. Genetic engineering has been very successful in bioremediation processes [69]. For example, in metal-contaminated environments, microbes

have been subjected to the introduction of the bmtA gene encoding for metallothionein, transforming it into a vector suitable for bioremediation. *Pseudoalteromonas haloplanktis* was also used for the same purpose. It possesses a shuttle plasmid encoding the suppressor for amber mutation. Other bacteria equipped with a plasmid in which the merA gene is present that enables the transformation of mercury in its toxic form into its nontoxic form can be converted into marine bacteria for mercury bioremediation. Finally, *Deinococcus radiodurans* has also been genetically modified and is the most radio-resistant organism. It has been made suitable for consumption and for the digestion of toluene and the ionic form of mercury from nuclear waste.

### 5.6. Use of Nanomaterials for Marine Bioremediation

Due to the high concentration of pollutants and xenobiotics or refractory compounds, it appears that bioremediation is not entirely effective, even though it provides an excellent and flexible recovery strategy. This is because it can cause unsustainable treatment efficiencies and recovery times [92]. In this context, the development of nanotechnology, which is one of the most advanced technologies and arose from the synergy of physics, chemistry, and biology producing various types of materials including nanoparticles (NPs) and nanomaterials [93], represents a promising innovation in the enhancement of the process of bioremediation and overcomes the limitations for in situ or ex situ application [94]. Depollution of hazardous and radioactive waste pollution, groundwater and wastewater treatment, and heavy metal and hydrocarbon-contaminated sediments remediation are part of the most known potential applications of nanomaterials (NMs) in bioremediation processes [95]. Nanoparticles such as Au, Cu, FeNi, Cu3Au, carbon-based nanomaterials (nanotubes), metal oxide NPs, nanocomposites and bionanomaterials (e.g., viruses, plasmids, and protein NPs) are some examples of nanomaterials actually used. In addition to directly catalyzing the degradation of pollutants, nanomaterials promote the development of micro-organisms which are able to degrade toxicants. Multiwall carbon nanotubes (MWCNT) designed to absorb oil and heavy metal hydrocarbons represent a good example and were applied to support Antarctic oil-degrading microbial flora [96].

### 6. Conclusions

Most aquatic pollutants are synthetic so their degradation or removal is not always easy. The impact of pollution on the ecology of an aquatic system can lead to multiple consequences that reflect the habits, biology, and ecology of the species. In addition to their negative biological impact, biological pollutants can spread a range of diseases in the water. Nowadays, for public and environmental health, attention must be paid to assessing the quality of aquatic ecosystems through a series of controls and regulations to be followed with strict deadlines. However, the pollution of aquatic systems worldwide is continuously increasing due to society today. Fortunately, there is a growing interest in the use of biodegradation, an innovative technique that harnesses the enzymatic activities of micro-organisms and the efficiency of nanomaterials as sustainable ways to clean up compromised environments.

Unfortunately, marine and coastal ecosystems are among the most threatened in the world, and this is no insignificant problem as they are also among the most productive. Recent approaches such as molecular ecology, metagenomics, and ecological modelling are important for evaluating and implementing the microbial bioremediation process. The high microbial diversity in aquatic ecosystems provides an abundance of untapped genetic information, bioactive compounds, and biomaterials with potential applications which are of societal interest. The innovation provided by the bioremediation process translates into ensuring favorable conditions for the degradation of contaminants using micro-organisms already present in nature and, possibly, the introduction of specific species for different toxic compounds.

Prevention techniques should be improved. Indeed, prevention should be the main solution because some of the effects of pollution can also be irreversible, causing permanent

damage to marine ecosystems. The combination of legislation and ecological awareness-raising through careful advocacy would be the most appropriate method, and it would be the responsibility of both the public and the scientific community to ensure that politics and businesses pay more attention to this problem. Environmental hazards that threaten ecosystems, such as pollution by plastics, metals, and xenobiotics, must be very quickly addressed. It is crucial because the problem affects the environment, marine organisms, and consequently, humans who feed on aquatic organisms, providing a pathway for toxins to enter, leading to a range of consequences including cancer, problems in children, and possible long-term illnesses. "An ounce of prevention is worth a pound of cure". Man is in close contact with the marine environment and depends on it. If we are to continue to depend on the aquatic environment, more attention is needed from everyone.

**Author Contributions:** Conceptualization, V.A.; methodology, V.A. and C.R.M.; writing—original draft preparation, V.A. and C.R.M.; writing—review and editing, V.A., C.R.M., B.T. and C.F.; preparation V.A., C.R.M. and C.F.; supervision, C.F.; project administration, V.A. and C.F. All authors have read and agreed to the published version of the manuscript.

**Funding:** This research received no external funding.

**Institutional Review Board Statement:** Not applicable.

**Informed Consent Statement:** Not applicable.

**Data Availability Statement:** Not applicable.

**Conflicts of Interest:** The authors declare no conflict of interest.

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
