# Peer review of "Get Rid of Marine Pollution: Bioremediation an Innovative, Attractive, and Successful Cleaning Strategy"

_sustainability, doi:10.3390/su141811784_

Round 1

Reviewer 1 Report

In this overview, the authors outline the most common problems of chemical pollution in marine environment and directions of the utilization of foreign substances with the assistance of living organisms. This point of view is very useful and gives an opportunity to deeper inside in each of mentioned items.

To give such opportunity, I recommend to provide more references to general or basic works within a field in each paragraph, because the presented references are mostly devoted to the particular mechanistic studies.

I also have some specific comments concerning the utilized definitions (xenobiotics) and organization of some paragraphs.

Specific comments (L –means Line)

 Keywords:  monitoring.Monitoring was not discussed here. Reject, please

L 31 refers to an aquatic environment – you mean marine environment, not freshwater

 a high rate of pollutants’  - it can be not high but combination of plural pollutants that can be toxic iven as micropollutants

The definition of (https://education.nationalgeographic.org/resource/marine-pollution) is more precise: ‘Marine pollution is a combination of chemicals and trash, most of which comes from land sources and is washed or blown into the ocean’. This review is concentrated on the chemical pollution. Besides, when we say about micropollutants, a low rate can also be dangerous in their combination.

L 35 Concerning marine pollution ‘atmosphere plays a key role’ ‘it  is not correct:… and what about a storm waters, destroyed purification plants?

Please , provide a references to 1 and 2 paragraphs

L85-86 – you mean ‘other xenobiotics’ – but both nonessential, mostly Heavy metals and radioactive substances are also xenobitics when they input the organism

L 90 The elimination of heavy metals at the tissue level is not constant. specify, please

L 94 please, provide references.

L 110 -  you mean ‘macrofibres’ or micro…?

L 132 ‘transported through the atmosphere. – not only: ‘Point sources of insecticides include wastewater treatment facilities (which receive runoff during non-overflow conditions), combined sewer overflows (CSOs) during wet weather, manufacturing facilities and insecticide spills and leaks on farms or other areas where they are stored and handled in bulk quantities. This would include back-siphoning of insecticides into irrigation water wells not equipped with proper safety devices. Also, agricultural ditches that convey runoff or irrigation returns may act as point sources. Nonpoint sources of insecticides are less spatially localized. Insecticides typically enter waterbodies with surface water runoff. (https://www.epa.gov/caddis-vol2/insecticides);

Agricultural insecticides threaten surface waters at the global scale. Sebastian Stehle and Ralf Schulz  https://doi.org/10.1073/pnas.1500232112

L 136. Why you include PPCPs in the paragraph concerning pesticides? (Ebele et al., 2017; https://doi.org/10.1016/j.emcon.2016.12.004)

3. The effects. Please, give some statements concerning the environmental suitability of these units for the assessment of the effects. How withstand the titles 2 (Toxic substances and their effects) and 3 (The effects)? The 3 is the part of 2 according to these titles.

L 201 – not only LD50.’ And better not effects of toxicity’’ but the severity of toxicity.

4. L 205 ‘that require attention’ – unclear,  in what sense they require attention?

L 201 – very scant characteristic. Precisely LC, not LD for aquatic organisms, and also the ratio of an acute toxicity test result (ie. LC50) to a chronic toxicity test result (ie. NOEC); The LC50 should be expressed as a time-dependent value (eg. 24-h or 96-h LC50).; Lowest Observed Effect Concentration (LOEC) It is also called lowest observed adverse effect level (LOAEL);  Maximum Acceptable Toxicant Concentration (MATC).

L 215 – In my opinion, endocrine disruptors are here in a wrong place. They can be combined with PPCPs.

Concerning biomagnification, the examples for toxic metals and hydrophobic substances are more suitable. In this paragraph, only one reference is given. Please, add the units Bioconcentration Factor (BCF) or Bioaccumulation Factor (BAF) 

It is better to determine Bioaccumulation, bioconcentration, biomagnification’’ (Alexander, D.E. (1999). Bioaccumulation, bioconcentration, biomagnification. In: Environmental Geology. Encyclopedia of Earth Science. Springer, Dordrecht. https://doi.org/10.1007/1-4020-4494-1_31).

Some examples can be also given:

Kamel Boudjema, Abdelmalek Badis, Nadji Moulai-Mostefa, Study of heavy metal bioaccumulation in Mytilus galloprovincialis (Lamark 1819) from heavy metal mixture using the CCF design, Environmental Technology & Innovation, Volume 25, 2022,

Assessment of the heavy metals contamination in bivalve Mytilus galloprovincialis using accumulation factors. : Rosioru, D. M. ;  Oros, A. ;  Lazar, L. Journal of Environmental Protection and Ecology 2016 Vol.17 No.3 pp.874-884 ref.25

Gobas, F.A.P.C. (2001). Assessing Bioaccumulation Factors of Persistent Organic Pollutants in Aquatic Food-Chains. In: Harrad, S. (eds) Persistent Organic Pollutants. Springer, Boston, MA. https://doi.org/10.1007/978-1-4615-1571-5_6

Schäfer S, Buchmeier G, Claus E, Duester L, Heininger P, Körner A, Mayer P, Paschke A, Rauert C, Reifferscheid G, Rüdel H, Schlechtriem C, Schröter-Kermani C, Schudoma D, Smedes F, Steffen D, Vietoris F. Bioaccumulation in aquatic systems: methodological approaches, monitoring and assessment. Environ Sci Eur. 2015;27(1):5. doi: 10.1186/s12302-014-0036-z.

6.1. Biodegradation of Petroleum Hydrocarbons and 6.2 Biodegradation of Xenobiotics. 6.3.

Incorrect titles – because 6.1. and toxic metals (Cd, Pd) are xenobiotics for the marine organisms.

Author Response

In this overview, the authors outline the most common problems of chemical pollution in marine environment and directions of the utilization of foreign substances with the assistance of living organisms. This point of view is very useful and gives an opportunity to deeper inside in each of mentioned items. 

To give such opportunity, I recommend to provide more references to general or basic works within a field in each paragraph, because the presented references are mostly devoted to the particular mechanistic studies.

I also have some specific comments concerning the utilized definitions (xenobiotics) and organization of some paragraphs.

We are grateful to the reviewer for the general comments and for providing us with detailed comments to improve the review. We followed all the points, trying to improve the quality and the clarity of the work, considering the reviewer’s suggestion.  

Specific comments (L –means Line)

Keywords:  monitoring.Monitoring was not discussed here. Reject, please. Done.

L 31 refers to an aquatic environment – you mean marine environment, not freshwater‘

a high rate of pollutants’  - it can be not high but combination of plural pollutants that can be toxic iven as micropollutants. Done.

The definition of (https://education.nationalgeographic.org/resource/marine-pollution) is more precise: ‘Marine pollution is a combination of chemicals and trash, most of which comes from land sources and is washed or blown into the ocean’. This review is concentrated on the chemical pollution. Besides, when we say about micropollutants, a low rate can also be dangerous in their combination. Thank you, corrected.

L 35 Concerning marine pollution ‘atmosphere plays a key role’ ‘it is not correct:… and what about a storm waters, destroyed purification plants? The authors improved this part.

Please , provide a references to 1 and 2 paragraphs. Done.

L85-86 – you mean ‘other xenobiotics’ – but both nonessential, mostly Heavy metals and radioactive substances are also xenobiotics when they input the organism. Corrected.

L 90 ‘The elimination of heavy metals at the tissue level is not constant.’ specify, please – The following phrase explains the reason why the elimination is not constant, i.e., because it depends on a series of elements.

L 94 please, provide references. Done.

L 110 -  you mean ‘macrofibres’ or micro…? Corrected.

L 132 ‘transported through the atmosphere. – not only: ‘Point sources of insecticides include wastewater treatment facilities (which receive runoff during non-overflow conditions), combined sewer overflows (CSOs) during wet weather, manufacturing facilities and insecticide spills and leaks on farms or other areas where they are stored and handled in bulk quantities. This would include back-siphoning of insecticides into irrigation water wells not equipped with proper safety devices. Also, agricultural ditches that convey runoff or irrigation returns may act as point sources. Nonpoint sources of insecticides are less spatially localized. Insecticides typically enter waterbodies with surface water runoff.’ (https://www.epa.gov/caddis-vol2/insecticides); Thank you for the suggested paper, I added relevant information.

Agricultural insecticides threaten surface waters at the global scale. Sebastian Stehle and Ralf Schulz https://doi.org/10.1073/pnas.1500232112

L 136. Why you include PPCPs in the paragraph concerning pesticides? (Ebele et al., 2017; https://doi.org/10.1016/j.emcon.2016.12.004) A new paragraph explaining the action of antimicrobial compounds was created after the “Pesticides” one.

  1. The effects. Please, give some statements concerning the environmental suitability of these units for the assessment of the effects. How withstand the titles 2 (Toxic substances and their effects) and 3 (The effects)? The 3 is the part of 2 according to these titles. We decide to delete the paragraph.

L 201 – not only LD50.’ And better not effects of toxicity’’ but the severity of toxicity. 

  1. L 205 ‘that require attention’ – unclear,  in what sense they require attention? The authors would like to say that it is important to understand how these two processes occur. However, I modify the phrase to make it clearer.

L 201 – very scant characteristic. Precisely LC, not LD for aquatic organisms, and also the ratio of an acute toxicity test result (ie. LC50) to a chronic toxicity test result (ie. NOEC); The LC50 should be expressed as a time-dependent value (eg. 24-h or 96-h LC50).; Lowest Observed Effect Concentration (LOEC) It is also called lowest observed adverse effect level (LOAEL);  Maximum Acceptable Toxicant Concentration (MATC). Thank you for the explanation, but we decide to delete the paragraph.

L 215 – In my opinion, endocrine disruptors are here in a wrong place. They can be combined with PPCPs. The paragraphs were remodulated. The paragraph was remodulated, moving the section about the “endocrine-disruptors” in the “antimicrobial” paragraph.

Concerning biomagnification, the examples for toxic metals and hydrophobic substances are more suitable. In this paragraph, only one reference is given. Please, add the units Bioconcentration Factor (BCF) or Bioaccumulation Factor (BAF) Done.

It is better to determine Bioaccumulation, bioconcentration, biomagnification’’ Done. (Alexander, D.E. (1999). Bioaccumulation, bioconcentration, biomagnification. In: Environmental Geology. Encyclopedia of Earth Science. Springer, Dordrecht. https://doi.org/10.1007/1-4020-4494-1_31). Done, and thank you for the provided papers.

Some examples can be also given: 

Kamel Boudjema, Abdelmalek Badis, Nadji Moulai-Mostefa, Study of heavy metal bioaccumulation in Mytilus galloprovincialis (Lamark 1819) from heavy metal mixture using the CCF design, Environmental Technology & Innovation, Volume 25, 2022,

Assessment of the heavy metals contamination in bivalve Mytilus galloprovincialis using accumulation factors. : Rosioru, D. M. ;  Oros, A. ;  Lazar, L. Journal of Environmental Protection and Ecology 2016 Vol.17 No.3 pp.874-884 ref.25

Gobas, F.A.P.C. (2001). Assessing Bioaccumulation Factors of Persistent Organic Pollutants in Aquatic Food-Chains. In: Harrad, S. (eds) Persistent Organic Pollutants. Springer, Boston, MA. https://doi.org/10.1007/978-1-4615-1571-5_6

Schäfer S, Buchmeier G, Claus E, Duester L, Heininger P, Körner A, Mayer P, Paschke A, Rauert C, Reifferscheid G, Rüdel H, Schlechtriem C, Schröter-Kermani C, Schudoma D, Smedes F, Steffen D, Vietoris F. Bioaccumulation in aquatic systems: methodological approaches, monitoring and assessment. Environ Sci Eur. 2015;27(1):5. doi: 10.1186/s12302-014-0036-z. 

6.1. Biodegradation of Petroleum Hydrocarbons and 6.2 Biodegradation of Xenobiotics. 6.3.

Incorrect titles – because 6.1. and toxic metals (Cd, Pd) are xenobiotics for the marine organisms. The paragraphs were modulated by deleting “biodegradation of xenobiotics”

Reviewer 2 Report

Dear Authors,

The manuscript is well written. There are few minor corrections which are highlighted in the attached PDF file. Kindly do the needful.

Author Response

We are grateful to the reviewer for the positive general comment and for giving us such relevant suggestions. We followed all the points, modifying the manuscript as required.

Reviewer 3 Report

The manuscript entitled” get rid of marine pollution bioremediation and innovation attractive and successful cleaning strategy” is under the scope of the Journal. The manuscript is well written and covers most of the aspects as per the title. However, the manuscript needs revision before final acceptance. There are some comments below.

Comments:

·         Revised the abstract, which must cover the manuscript.

·         Page 2, instead of bullet form use the number

·         Authors must add their opinion last paragraph of each section

·         The weakest point of the manuscript is the lake of tables, figures, and schematic diagrams. Authors must add all the above which will improve the quality of the manuscript.

·         Section 2.2, discussed the size of microplastics and their origin and also cites some recent works

·         Section 2.5 and 3 same as above, instead of using bullet form use number form for the main water-borne pollutants are

·         Improve and cite some recent works in section 3

·         Section    6.1 add schematic diagram

·         Improve the conclusions section

·         The references are not as per journal guidelines, authors must check them one by one reference.

Author Response

The manuscript entitled” get rid of marine pollution bioremediation and innovation attractive and successful cleaning strategy” is under the scope of the Journal. The manuscript is well written and covers most of the aspects as per the title. However, the manuscript needs revision before final acceptance. There are some comments below.

We are grateful to the reviewer for the general and specific comments. We followed all the reviewer’s points, in order to improve the manuscript.

Comments:

  • Revised the abstract, which must cover the manuscript. The abstract has been edited.
  • Page 2, instead of bullet form use the number. We deleted the list of phrases and modified the text making it more smooth.
  • Authors must add their opinion last paragraph of each section. Done.
  • The weakest point of the manuscript is the lake of tables, figures, and schematic diagrams. Authors must add all the above which will improve the quality of the manuscript. Done.
  • Section 2.2, discussed the size of microplastics and their origin and also cites some recent works. Done.
  • Section 2.5 and 3 same as above, instead of using bullet form use number form for the main water-borne pollutants are. Done.
  • Improve and cite some recent works in section 3. We decide to delete the paragraph
  • Section    6.1 add schematic diagram. Done.
  • Improve the conclusions section. Done.
  • The references are not as per journal guidelines, authors must check them one by one reference. Done.

Round 2

Reviewer 2 Report

Dear Author

The suggested corrections are incorporated in the manuscript. Figure 2 requires sharpness. So do that correction.

Reviewer 3 Report

Dear Editor

Good day

The authors have addressed all the comments as per requested, now the manuscript is ready for final acceptance.

Regards

Faiz